# Influence of Chemical Pretreatment on the Adsorption of N$_2$ and O$_2$ in Ca-Clinoptilolite

Miguel Ángel Hernández [1,*], Gabriela I. Hernández [2], Roberto Ignacio Portillo [3], Ma de los Ángeles Velasco [1], Juana Deisy Santamaría-Juárez [4], Efraín Rubio [5] and Vitalii Petranovskii [6]

[1] Departament of Zeolites Research, Posgraduate in Agroecology, ICUAP, Meritorious Autonomous University of Puebla, Puebla 72570, Mexico

[2] Department of Process Engineering, Metropolitan Autonomous University-Iztapalapa, Ciudad de Mexico 09340, Mexico

[3] Faculty of Chemical Sciences, Meritorious Autonomous University of Puebla, Puebla 72570, Mexico

[4] Faculty of Chemical Engineering, Meritorious Autonomous University of Puebla, Puebla 72570, Mexico

[5] University Center for Linking and Technology Transfer, Meritorious Autonomous University of Puebla, Puebla 72570, Mexico

[6] Nanosciences and Nanotechnology Center, UNAM, Ensenada 22890, Mexico

[*] Correspondence: miguel.hernandez@correo.buap.mx; Tel.: +54-222-229-5500 (ext. 7270)

**Abstract:** N$_2$ and O$_2$ adsorption isotherms in chemically modified clinoptilolite-Ca zeolites were experimentally estimated by inverse adsorption chromatography. Natural zeolites (CLINA) were chemically treated with HCl at different concentrations (H1-H4). The adsorption of N$_2$ and O$_2$ on these zeolites was studied in the temperature zone of 398–498 K using gas chromatography. This technique used a thermal-conductivity detector and He as carrier gas, at a rate of 30 mL min$^{-1}$. The Langmuir and Henry equations were used to describe the experimental results of these gases' adsorption. To evaluate the selectivity of the components of atmospheric air, the chemical activation of the zeolite clinoptilolite-Ca has been carried out. The results are attractive because of the ability to separate the gases these nanomaterials present under dynamic conditions. The structural modifications of the crystalline phases of the studied zeolites were carried out through X-ray diffraction, where the average crystal size was evaluated with the Scherrer equation, finding values of 25.86 nm for CLINA and 15.12 nm for H3 zeolites. The variation of their chemical composition was carried out by energy-dispersive EDS, while the adsorption of N$_2$ carried out their texture properties at 77 K. The selectivity coefficients ($\alpha$) were evaluated for these gases in pure form and in a mixture (atmospheric air), finding the highest values in zeolites H4 and H3. The interaction energies between these gases with the porous structures of the studied zeolites were evaluated from the evolution of the isosteric enthalpies of adsorption through the Clausius–Clapeyron equation.

**Keywords:** clinoptilolite; zeolite; N$_2$; O$_2$; adsorption; low coverage; chromatography

## 1. Introduction

Air separation is one of the most important gas-separation processes in the modern chemical industry. Cryogenic distillation and adsorption processes are two of the most common methods used for the separation of O$_2$ and N$_2$ from air. Examples of the latter include pressure swing adsorption (PSA). This technique (PSA) works based on the selective adsorption-capability of a component on a suitable adsorbent. Usually, finding a suitable adsorbent is the first step in developing an adsorption separation process [1]. The separation factor ($\alpha$) generally varies with temperature and composition. Therefore, choosing appropriate conditions for maximizing the separation factor is a major issue in the design process [2]. Occasionally, an initial selection of suitable adsorbents can be directly achieved according to the available Henry's law constants. Most of the time, selecting the adsorbents can be easily achieved with measurement of chromatographic retention times

(RT) [3]. Chromatographic methods are better for achieving a quick and actual estimation of separation factors, because of their prediction in adsorption kinetics. Synthetic zeolites 5A, 13X and natural small-pore zeolites such as CHA, ERI, MOR and HEU clinoptilolites, are the most used adsorbents in air separation for $O_2$ production [4]. Of these natural zeolites, one of the most used is clinoptilolite (the structural code is HEU, according to the database of the International Zeolite Association, IZA) [5]. Clinoptilolite is a hydrophilic zeolite and is characterized by a two-dimensional system of channels of three types. These are two parallel channels, channel A (0.72–0.44 nm) and channel B (8-member ring, 0.41–0.47 nm), which are perpendicularly intersected by channel C (8-member ring, 0.55–0.40 nm). In the clinoptilolite structure, shown in Figure 1, the channels are filled with water molecules and exchangeable cations.

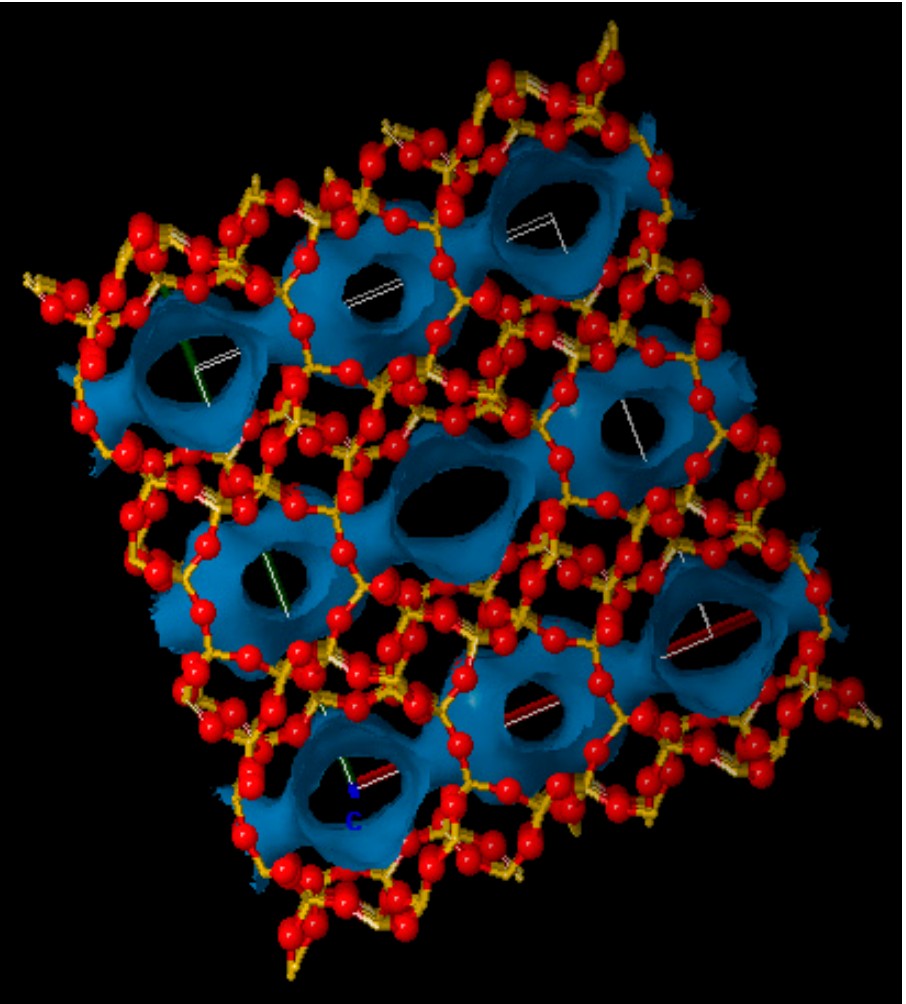

**Figure 1.** A three-dimensional simulation that schematically shows Si and O ions (yellow and red balls, respectively) and the internal surface of channels (light-blue surface) [5].

These cations are in the main sites of the clinoptilolite, and they are coordinated with different numbers of water molecules and oxygen atoms. Some of the factors that affect the thermal and adsorptive behavior of these zeolites are the molar ratio, Si/Al, the ionic potential, exchangeable-cations size and composition, impurities, their coordination after water expulsion and framework topology. For adsorbed molecules, the size of the input windows formed by oxygen bridges determines the accessibility of the internal cavities of zeolites. Replacing the type of exchangeable cations can change the size of windows [6]. Natural zeolites are complex adsorption systems, since they possess micropores (pore diameter, W < 2 nm), mesopores (W = 2–50 nm), and macropores (W > 50 nm) [7]. During

the adsorption process, the initial filling of micropores is accompanied by a gradual surface coverage of the walls of the open mesopores [8]. Secondly, the macropores are covered by adsorbate molecules [8]. This process occurs at higher pressures, and comprises monolayer, multilayer, and capillary condensation [8]. When the adsorption occurs inside micropores smaller than twice the diameter of the adsorbate molecule there is an enhancement of the adsorption energy. This enhancement occurs due to the overlap between potential fields generated on both sides of the pore walls. Under these circumstances, there is a mechanism designated as micropore volume-filling [9,10]. Several solids with pores of molecular dimensions (micropores) are often used as selective adsorbents in industrial applications, due to this enhanced-adsorption attribute.

Within this subject, nanoporous materials play a significant role. For example, natural and synthetic zeolites, are remarkable for their uses in different fields of contemporary technology. Registered clinoptilolite deposits can be found in a variety of places in Mexico [11]. Samples from these deposits were selected for a detailed study. Clinoptilolite in various cationic forms is the main phase of these rocks. However, as in any natural material, there is a certain number of impurities in their composition. In the description of porous materials, it is useful to consider the concepts of primary and secondary porous structures [12]. The primary porosity of zeolites depends on the crystal structure and composition of the zeolite. Usually, this surface is in the range of several hundred square-meters per gram of zeolite. The external surface area, known as mesopores, contributes a small percentage of the total available area, Figure 2.

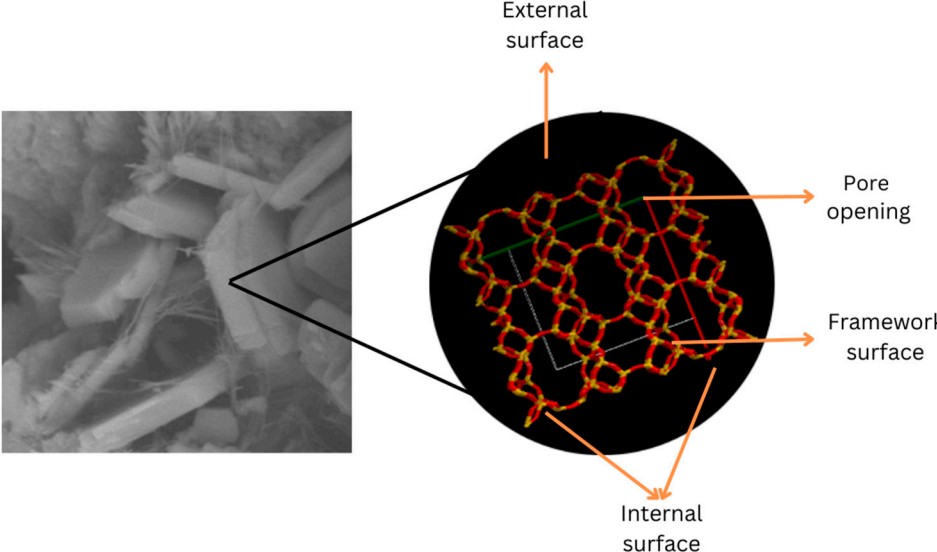

**Figure 2.** An HEU zeolite crystal, channel structures and external surface (exemplified by the (010) surface). The channels are drawn superimposed onto the crystals for illustration purposes and are not drawn according to scale. The external surface consists of pore openings and the framework surface.

The nanoporosity of these materials can be evaluated using high-resolution adsorption (HRADS) studies of molecules of diverse sizes, as well as by other methods [13]. The use of natural zeolites is truly diverse, but their universality is discussed with reference to their synthetic analogs. It has been reported [14,15] that natural sedimentary zeolite cannot replace synthetic zeolites in some important industrial applications, because they have a limited chemical purity and multiphase composition. Nevertheless, other authors note there are already numerous applications of natural zeolites, especially clinoptilolite. Nonetheless, despite the increasing use of zeolites, their use is gradually becoming more complex, due to the limitations imposed by the size of micropores. In this sense, to overcome the diffusion limitations, the modern strategy of obtaining synthetic materials consists in creating shorter diffusion paths [16,17]. This characteristic is obtained by creating mesopores, in addition to internal microporosity [16,17]. Improvements obtained in this respect have aroused

great interest in the use of zeolites. These enhancements demonstrate the existence of a hierarchical structure. These structures possess a mesopore system in addition to nanopores. The discovery that natural zeolites can operate in many unexpected sectors has expanded the scope of their application, opening new horizons. In this work, we investigate and compare the adsorption of $O_2$ and $N_2$ in a series of zeolites of framework-type HEU. Additionally, we evaluate the degree of selectivity ($\alpha$) of these gases. The zeolites used were natural, as well as chemically treated with HCl at different concentrations. The novelty of this work is based on the study of the nanoporosity of the HEU-type zeolites through adsorption studies, using their applications in the separation of $O_2$ and $N_2$ by inverse adsorption chromatography IGC [18]. The aim of this work is to use the degree of selectivity ($\alpha$) and isosteric enthalpy of adsorption ($q_{st}$) to estimate the degree of interaction of $O_2$ and $N_2$ in natural clinoptilolite zeolite (CLINA) and decationated with HCl (H1-H4) in low concentrations.

## 2. Materials and Methods

### 2.1. Materials

　　Reactives HCl and $AgNO_3$ were purchased from Aldrich (J.T. Baker, Phillipsburg, NJ, USA). The gases $O_2$, $N_2$, and He > 99.999%, supplied by INFRA Corp., were used.

### 2.2. Methodology

Dealumination of Clinoptilolite

　　The clinoptilolite zeolites used in this work were collected in San Gabriel Chilac, Puebla, Mexico. The adsorbents were natural zeolite (CLINA) and selected dealuminated clinoptilolite-zeolites (H1, H2, H3, and H4). The CLINA label accounts for the natural sample, which is free of any chemical treatment. Samples of dealuminated clinoptilolite-zeolites were prepared at laboratory scale by means of an acid-leaching process of the CLINA precursor. This leaching process caused the removal of impurities and the exchange of polyvalent cations by protons. This modification procedure consisted of a series of washing cycles of natural clinoptilolite samples (mesh 60–80) with dilute HCl. Secondly, they were subject to prolonged rinses with deionized water, which was confirmed with $AgNO_3$ as indicator. Each of the steps of the HCl washing procedure of the sample was made at 50 °C for 6 h, with 0.01 N HCl. A sample labeled as H1 was prepared from CLINA, after a sample of the natural zeolite was treated once with HCl (H1 labeling follows the number of washings with HCl). Likewise, a sample branded as H2 ensued from CLINA after treating a sample of the natural zeolite twice, with HCl.

### 2.3. Experimental Measurement Techniques

　　(i) *XRD.* To identify the crystalline phases, the X-ray powder-diffraction (XRD) technique was used, using a Bruker model D8 Discover diffractometer (Bruker, Co., Billerica, MA, USA) employing nickel-filtered Cu K$\alpha$ ($\lambda$ = 0.154 nm) radiation, operated at 40 kV and 30 mA. The samples were pulverized in an agate mortar and placed in a sample holder, compressing the powder until it was perfectly compact. The crystalline phases were identified using the 2013 PDF4+ database of the International Center for Diffraction Data (ICDD). The mean average-crystalline-size of zeolites was calculated using the Scherrer equation [19].

　　(ii) *SEM.* Images of the zeolite samples in the study were obtained with a JEOL, model JSM-7800F (JEOL USA, Inc., Peabody, MA, USA) high-resolution scanning electron microscope at 5 kV. The samples were mounted on aluminum stub-holders and subsequently coated with Au, using a sputtering coater.

　　(iii) *EDS.* An elemental chemical analysis using X-ray energy dispersion (EDS) was performed on each of the zeolites, using an Oxford INCA energy 250+ model probe, with a resolution of 137 eV and a 10 mm$^2$ detector.

　　(iv) *$N_2$ Adsorption.* All $N_2$ adsorption–desorption isotherms were measured at the boiling point of liquid $N_2$ (76.7 K at the 2200-m altitude of Mexico City) using automatic

volumetric-adsorption system (Quantachrome AutoSorb-1C, Quantachrome Instruments, Boynton Beach, FL, USA). $N_2$ adsorption isotherms were determined in the interval of relative pressures, $p/p^0 = 10^{-6}$ to 0.995. The saturation pressure, $p^0$, was continuously registered during the adsorption–desorption measurements. Powder-particle sizes corresponding to mesh 60–80 were sampled from all zeolites under analysis. Prior to the sorption experiments, samples of zeolites were outgassed at 623 K for 15 h at a pressure less than $10^{-6}$ Torr. The relevant equations used for the calculation of the surfaces were The Brunauer–Emmett–Teller (BET), Langmuir, and t-plots (external) in the range of $p/p^0$ from 0.01 to 0.25. The total pore volume, $V_\Sigma$, was estimated using the Gurvitsch rule, based on the volume adsorbed at the relative pressure $p/p^0 = 0.95$, calculated as volume of liquid.

*2.4. Adsorption of $N_2$ and $O_2$*

Chromatographic $N_2$- and $O_2$-adsorption experiments were carried out in a GC-14A Shimadzu gas chromatograph (Shimadzu Co KK, Kyoto, Japan) equipped with a thermal conductivity detector. The chromatographic columns (i.d. = 5 mm, length = 50 cm) were made of glass and packed with zeolites (60–80 mesh). Adsorbate pulses of different intensities were injected into the incumbent GC column that was kept at a chosen temperature. The elution chromatogram of each pulse was registered until the recorder pen reached the baseline once more. Before the adsorption experiment, adsorbents were pretreated in situ under a flow of the He carrier gas at 573 K for 8 h. The gases, in pure form and in a mixture (atmospheric air) were injected separately, to measure their corresponding retention time inside the appropriate adsorbent column. The dynamic-adsorption method of gas chromatography or inverse adsorption chromatography, IGC, is the method followed in this work to obtain the chromatographic peaks. The isosteric enthalpy of adsorption of the $N_2$ and $O_2$ were calculated from sorption data evaluated at two temperatures through the Clausius–Clapeyron relation.

*2.5. Calculation Methods*

From gas-adsorption data at low pressures, it is possible to evaluate the Henry constants at different temperatures for the series of adsorbent–adsorptive pairs employed in this work, according to the following expression:

$$K_H = \lim_{p \to 0} \left( \frac{a}{a_m p} \right) \tag{1}$$

where $a$ represents the amount adsorbed on the solid walls at pressure $p$, while $a_m$ is the monolayer capacity evaluated from the Langmuir equation [20]:

$$\theta = \frac{a}{a_m} = \frac{kp}{1 + kp} \tag{2}$$

where is $kp = K_H$, something that can be tested graphically by plotting $1/a$ versus $1/p$:

$$\frac{1}{a} = \frac{1}{a_m} + \frac{1}{a_m kp} \tag{3}$$

$$\alpha_0 = \frac{(K_H\ N_2)}{(K_H O_2)} \tag{4}$$

where $K_H = (a_m)(K_L)$; $a_m$ is the Langmuir adsorption capacity and $K_L$ the constant from the Langmuir equation.

The selectivity coefficients of $N_2$ and $O_2$ can be evaluated by means of the Lewis equation [21]:

$$\alpha_1 = \frac{\left[ \frac{Y_1}{X_1} \right]}{\left[ \frac{Y2}{X2} \right]} \tag{5}$$

where $Y_1$ and $X_1$ are the mole fractions of the two adsorbates, *X*, *Y*, in the adsorbed phase and $X_2$ and $Y_2$ in the gas phase.

The chromatographic selectivity ($\alpha_2$) was analyzed by means of the relationship between the retention times (*tr*) and from the chromatographic-separation ratio ($\alpha_3$)

$$\alpha_2 = \frac{tr\ N_2}{tr\ O_2} \tag{6}$$

$$\alpha_3 = \frac{[2(tr\ N_2 tr O_2)]}{\frac{1}{2}\sum a} \tag{7}$$

Here, *tr* is the retention time of $N_2$ and $O_2$, and $\frac{1}{2}\sum a$ is the amplitude of the chromatographic peak measured at half their heights [22].

The selectivity coefficient based on the estimation of the isosteric enthalpy of adsorption ($\alpha_4$) was also evaluated, based on the following equation:

$$\alpha_4 = \left[\frac{qst\ N_2}{qst O_2}\right] \tag{8}$$

where *qst* is the isosteric enthalpy of adsorption, estimated from the Clausius–Clapeyron relation.

## 3. Results

Some important physical properties of the adsorptives employed in this work are listed in Table 1. From this table, the kinetic diameters ($\sigma$) of the adsorptive molecules are comparable to the sizes of the A–C channels of the clinoptilolite crystal structure.

**Table 1.** Physical constant of adsorbate gases [23].

| Gas | B. P. °C | Crit. Temp. °C | Quadrupole A³ | Polarizability A³ | Ion. Pot. Volts | Length Å | Width Å | σ-Kinetic Diameter, Å |
|---|---|---|---|---|---|---|---|---|
| $O_2$ | −183 | −118.8 | 0.10 | 1.2 | 12.5 | 2 | 1.4 | 3.46 |
| $N_2$ | −195.8 | −147.1 | 0.31 | 1.4 | 15.5 | 2.1 | 1.5 | 3.64 |

### 3.1. X-ray Analysis

In Figure 3, X-ray powder-diffraction of the CLINA-CLIDA zeolites studied are observed.

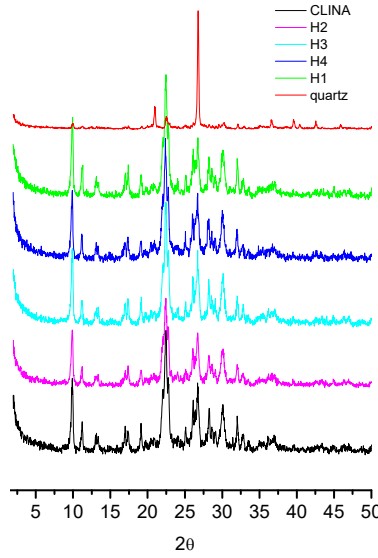

**Figure 3.** X-ray powder-diffraction pattern of natural and modified clinoptilolite.

In Table 2, the effect of chemical treatments can be observed, with the crystal size of the zeolites studied.

**Table 2.** Effect of chemical treatment with the size of crystal, XRD.

| Samples | Size of the Crystal nm |
|---|---|
| CLINA | 25.86 |
| H1 | 12.41 |
| H2 | 15.66 |
| H3 | 15.12 |
| H4 | 12.41 |

### 3.2. SEM

Scanning-electron-microscopic images of CLINA zeolite are given in Figure 4.

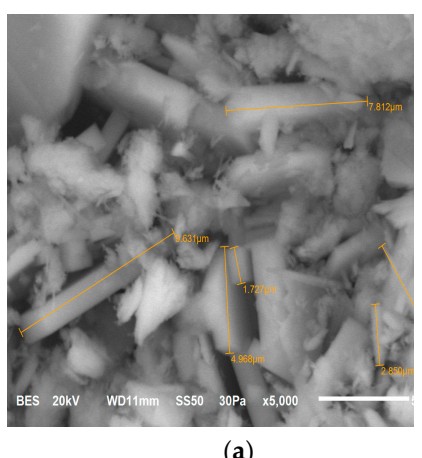
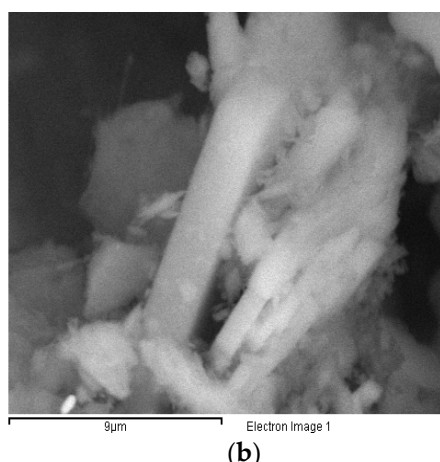

(**a**)  (**b**)

**Figure 4.** SEM micrographs of clinoptilolite zeolite CLINA: (**a**) ×5000; (**b**) ×1000.

### 3.3. The Chemical-Composition, EDS

The chemical-composition data estimated by EDS of the studied zeolites are given in Table 3.

**Table 3.** Chemical compositions of clinoptilolite zeolites, EDS (%).

| Sample | $SiO_2$ | $Al_2O_3$ | $Fe_2O_3$ | CaO | MgO | $Na_2O$ | $K_2O$ | $TiO_2$ | $CrO_2$ | Si/Al |
|---|---|---|---|---|---|---|---|---|---|---|
| CLINA | 67.07 | 11.31 | 1.21 | 3.75 | 0.68 | 2.90 | 0.52 | —— | —— | 5.22 |
| H1 | 38.59 | 9.63 | 6.51 | 0.853 | 1.84 | —— | 1.83 | —— | —— | 4.00 |
| H2 | 72.27 | 13.46 | 10.43 | 3.106 | 1.85 | 3.34 | 4.79 | 0.63 | —— | 5.36 |
| H3 | 40.27 | 7.66 | 4.57 | 3.106 | 1.85 | —— | 2.24 | 1.36 | —— | 5.25 |
| H4 | 82.95 | 8.99 | 2.59 | 0.783 | 1.42 | —— | 3.37 | 0.91 | 0.29 | 9.22 |

### 3.4. $N_2$ Adsorption

The $N_2$ adsorption isotherms at 77 K in the natural zeolite (CLINA) and in the chemically treated zeolites (H1-H4) are given in Figure 5, (relative pressure $p/p^0$ vs. adsorbed volume in $cm^3$ STP per gram of zeolite), and the experimental results are given in Table 4.

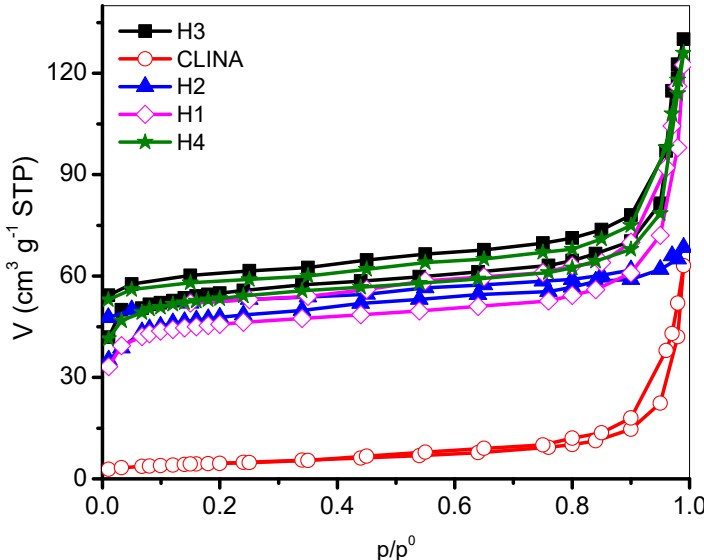

**Figure 5.** $N_2$ adsorption isotherms at 77 K on CLINA and H-modified zeolites.

**Table 4.** Textural parameters of natural (CLINA) and H-modified clinoptilolite zeolites.

| Zeolite | $As_L$ $m^2/g$ | $As_B$ $m^2/g$ | $C_B$ | $V_\Sigma$ $cm^3\ g^{-1}$ |
|---------|--------|--------|-------|-----------|
| CLINA | 27 | 18 | 201 | 0.035 |
| H1 | 32 | 21 | 147 | 0.084 |
| H2 | 62 | 46 | −200 | 0.105 |
| H3 | 149 | 97 | −51 | 0.153 |
| H4 | 183 | 120 | −45 | 0.154 |

$As_L$ is the specific Langmuir surface-area; $As_B$ is the specific BET surface-area; $C_B$ is the BET constant and $V_\Sigma$ is the volume adsorbed, close to saturation ($p/p^0 \sim 0.95$).

### 3.5. Adsorption of $O_2$, and $N_2$ on Clinoptilolite at Low Degrees of Coverage

Figure 6, a, a' and b, b', shows the experimental adsorption-isotherms obtained by the chromatographic method at 298 and 403 K; the results are listed in Table 5.

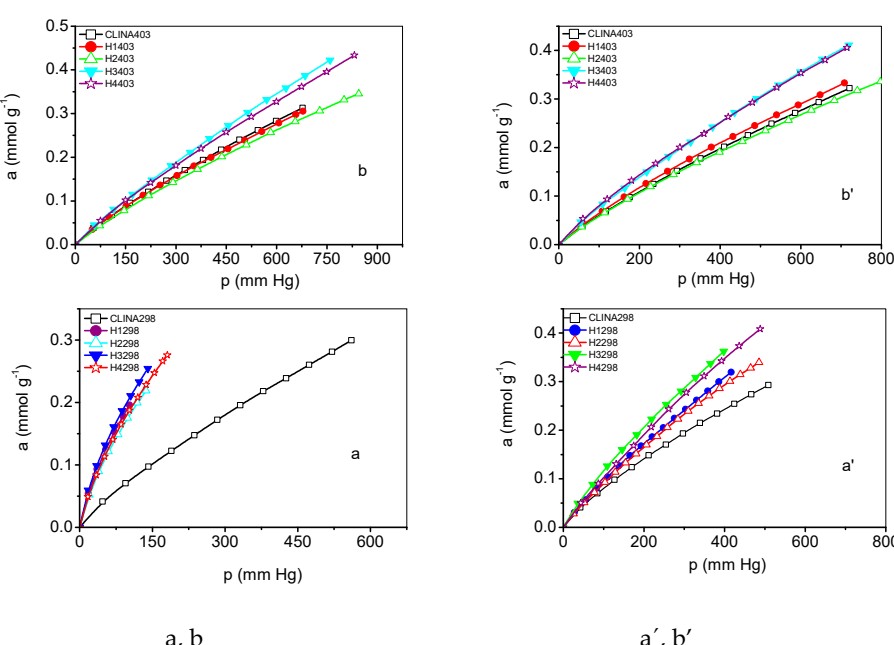

a, b                                        a′, b′

**Figure 6.** $N_2$ (**a**, **a'**) and $O_2$ (**b**, **b'**) adsorption isotherms at 298 K, and 403 K.

**Table 5.** Adsorption capacity ($a_m$, mmol/g), Langmuir constants ($K_L$, $10^3$ mm Hg$^{-1}$), Henry constants ($K_H$, $10^3$ mmol/g mm Hg) and selectivity coefficients ($\alpha_0$) in the zeolites studied.

| Sample | $a_m$ (N$_2$) | $a_m$(O$_2$) | KL(N2) | KL(O2) | KH(N2) | KH(O2) | $\alpha_0$ |
|--------|--------------|--------------|--------|--------|--------|--------|------------|
| H1 | 0.4799 | 0.7358 | 6.61 | 1.52 | 3.172 | 1.118 | 2.837 |
| H2 | 0.5416 | 0.668 | 4.87 | 1.61 | 2.637 | 1.075 | 2.453 |
| H3 | 0.5407 | 0.842 | 6.14 | 1.63 | 3.319 | 1.373 | 2.417 |
| H4 | 0.647 | 0.858 | 4.026 | 1.505 | 2.605 | 1.292 | 2.016 |

Table 5 lists the values corresponding to the adsorption capacity at T = 298 K and p = 100 mmHg, considering the molar fractions (X, Y) of both the adsorbed phase (capacity of the adsorption component) and the adsorbate [24].

The chromatographic separation ($\alpha_2$) of the mixture of gases in atmospheric air was analyzed in similar experimental conditions to the adsorption of pure gases in the zeolites studied, Table 6. In this table, the separation of these gases is achieved entirely in H4 zeolites. The relationship of the behavior of the retention times (tr) ($\alpha_3$) of the chromatographic peaks as a function of the acid treatment to which the studied zeolites are subjected, has also been analyzed. From this table, the tr corresponding to N$_2$ are initially sensitive to acid treatments (H1 and H2), and remain constant; in the case of tr for O$_2$, they remain practically constant. In this same table, for comparison, the values of $\alpha_0$ and $\alpha_1$ are listed.

**Table 6.** Adsorption capacity (mmol/g) and selectivity coefficient ($\alpha_1$) in the zeolites studied at T = 298 K; p = 100 mmHg; X$_1$ = 0.7 and X$_2$ = 0.3.

| Sample | N$_2$ | O$_2$ | Y$_1$ | Y$_2$ | $\alpha_1$ |
|--------|-------|-------|-------|-------|------------|
| H1 | 0.194 | 0.094 | 0.493 | 0.506 | 2.39 |
| H2 | 0.176 | 0.091 | 0.523 | 0.476 | 2.12 |
| H3 | 0.201 | 0.115 | 0.543 | 0.456 | 1.96 |
| H4 | 0.183 | 0.100 | 0.543 | 0.456 | 1.96 |

From this table, there are similarities between the separation coefficients. For example, from the selectivity coefficients of pure gases ($\alpha_0$), the simulated one ($\alpha_1$), are analogues to the evaluations from the mixture (atmospheric air) $\alpha_2$ and $\alpha_3$, although some inconsistencies occur, due to the diffusion process of the mixed gases and the size of the crystal.

*3.6. Isosteric Enthalpy of Adsorption and Selectivity Coefficients ($\alpha_4$)*

The behavior of the isosteric enthalpy of adsorption (-qst, kJ mol$^{-1}$) of N$_2$ and O$_2$ on the clinoptilolite zeolites is presented in Figure 7, and the results of these estimates are listed in Tables 7 and 8. These tables summarize the qst values for the gases used in this work, the net heat of adsorption (q$_N$) and the selectivity coefficients ($\alpha_4$) [25].

**Table 7.** N$_2$/O$_2$ gas selectivity-coefficients, pure and in mixture of the studied zeolites.

| Sample | $\alpha_0$ | $\alpha_1$ | $\alpha_2$ | $\alpha_3$ |
|--------|-----------|-----------|-----------|-----------|
| H1 | 2.837 | 2.39 | 2.864 | 2.111 |
| H2 | 2.453 | 2.12 | 2.58 | 2.769 |
| H3 | 2.417 | 1.96 | 2.42 | 2.182 |
| H4 | 2.016 | 1.96 | 2.36 | 2.20 |

**Table 8.** Isosteric enthalpies of adsorption (qst, Kcal/mol), net heat of adsorption (q$_N$, Kcal/mol) and selectivity coefficients ($\alpha_4$) for N$_2$ and O$_2$.

| Sample | q$_{st}$ (N$_2$) | q$_{st}$ (O$_2$) | q$_N$ (N$_2$) | q$_N$ (O$_2$) | $\alpha_4$ |
|--------|--------|--------|--------|--------|--------|
| N | 0.406 | 0.741 | −0.927 | −0.889 | 0.547 |
| H1 | 3.395 | 1.009 | 2.062 | −0.621 | 3.364 |
| H2 | 3.105 | 1.081 | 1.772 | −0.549 | 2.872 |
| H3 | 3.160 | 1.118 | 1.827 | −0.512 | 2.826 |
| H4 | 2.824 | 0.675 | 1.491 | −0.955 | 4.183 |

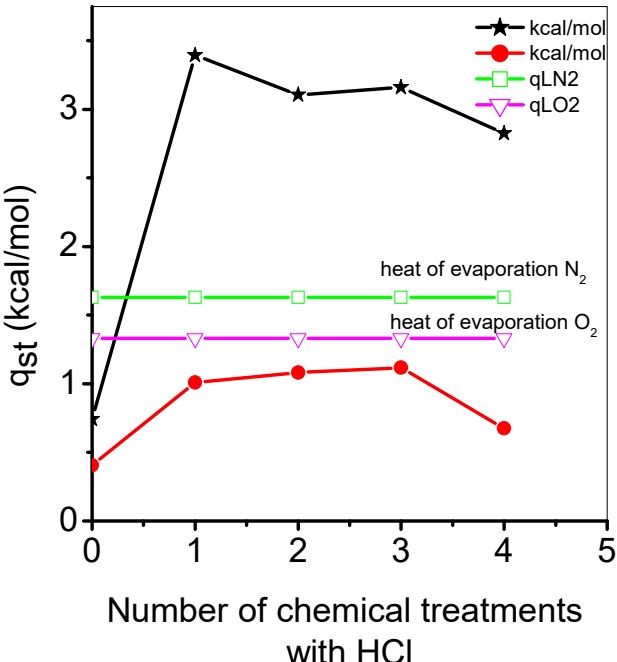

**Figure 7.** Isosteric enthalpies of adsorption (qst, Kcal/mol) for N$_2$ and O$_2$ in clinoptilolite zeolites.

## 4. Discussion

### 4.1. X-ray Analysis

From the diffraction pattern, there is no HCl leaching action, since it does not manifest substantial changes in the diffraction pattern of chemically modified zeolites, Figure 3. From these diffraction patterns, it is observed that the presence of all reflections corresponding to these zeolites is maintained: $2\theta = 9.85°, 11.08°, 13.03°, 14.84°, 16.86°, 17.02°, 19.04°, 20.73°, 22.35°, 23.88°, 25.42°, 26.24°, 27.00°, 28.09°, 30.01°, 32.31°, 32.57°$ and 34.80. As a comparison, a diffraction pattern of an $\alpha$-quartz ($2\theta \approx 27.5°$) [26] is introduced. The results shown indicate that the acid-modification procedure of zeolitic samples yielded an increase in the intensity of the peak corresponding to quartz. This behavior can be related to the progressive dealumination the crystalline structure suffered as the number of acid treatments increased, while quartz moieties remained mostly unaffected by the HCl leaching action. These dealuminated clinoptilolites zeolites may be difficult to study with diffraction techniques, and in this case NMR techniques may be the best available option [27]. From Table 2, it can be seen that the crystal size decreases as a function of the number of chemical treatments attributed to the chemical stability of these zeolites by the ratio of Si/Al.

### 4.2. EDS

The cation-blocking effects at pore entrances in natural clinoptilolites are thus diminished by acid treatment, thus lowering the cation-exchange capacity of the resultant substrates by leaching out Al$^{+3}$ from framework positions and introducing H$^+$ into the

remaining cation sites of the natural precursor [28]. From Table 2, the following sequence is observed for the chemically treated zeolites: Si: H4 > H2 > CLINA > H3 > H1; Al: H2 > CLINA > H1 > H4 > H3; Fe: H2 > H1 > H3 > H4 > CLINA; Ca: CLINA > H3 = H2 > H1 > H4; Mg: H2 = H3 > H1 > H4 > CLINA; Na: H2 > CLINA; K: H2 > H4 > H3 > H1 > CLINA; Ti: H3 > H4 > H2; Si/Al: H4 > H2 > H3 > CLINA > H1 and Si/Al: H4 > H2 > H3 > CLINA > H1. Acid treatment of these high-silica natural clinoptilolites has rendered improved adsorbents via the mechanism of decationation and dealumination, and by the dissolution of any amorphous silica blocking the channels A, B and C of the clinoptilolite structure.

### 4.3. SEM

The SEM of these zeolites shows us the form of the obtained glasses of the CLINA zeolite, presenting a cubic symmetry, with defined edges, Figure 4. From this figure it is observed that the crystals possess dimensions near to 2 μm.

### 4.4. Adsorption of $N_2$

Along the channels of this zeolite there exist ions (i.e., $Na^+$, $K^+$, $Ca^{2+}$, $Mg^{2+}$) that do not allow the passage of $N_2$ molecules into it. $N_2$ adsorption isotherms at 77 K on CLINA and H zeolites are shown in Figure 5. This figure portrays the evolution of the shapes of the $N_2$ isotherms with respect to the number of HCl washings. The CLINA and H1 substrates render IUPAC type IV isotherms (with extremely narrow hysteresis-loops), while a type I isotherm is proper for the H2 specimen (the hysteresis loop of this sample corresponds to an IUPAC type H4) [8]. Distinctive features of these types of isotherms are as follows:

(i) the extent of microporosity in dealuminated clinoptilolites increases, in general, with the number of acid treatments; the plateaus of the isotherms corresponding to H clinoptilolites reach increasing heights, according to the accessible microporosity depicted by each zeolite;

(ii) the existence of a low-pressure hysteresis region is evident for the H2 sample. Therefore, acid treatment of high-silica natural clinoptilolites can render adsorbents of enhanced accessible pore-volumes via the mechanism of decationation and dealumination, and by dissolution of any amorphous materials blocking the entrances to the A–C channels of the clinoptilolite structure. The cation-blocking effects at pore entrances in natural clinoptilolites are thus diminished by acid treatment. They lowered the cation-exchange capacity of the resultant substrates by leaching out $Al^{+3}$ from framework positions and introducing $H^+$ into the remaining cation sites of the natural precursor. Important textural parameters of CLINA and H clinoptilolites are listed in Table 3. For the H clinoptilolites, the BET equation $C$ constants are sometimes negative, and this can be explained by the fact that multilayer adsorption in micropores is not a plausible model therein. The isotherm of the CLINA zeolite shows an upward deviation at high relative pressures, due to multilayer formation and capillary condensation taking place in the mesopores (secondary porosity) of this sample.

### 4.5. Adsorption of $O_2$ and $N_2$ on Clinoptilolite at Low Degrees of Coverage

From Figure 6, we see that in the natural zeolite there is a slight increase in the adsorption capacity of the zeolites studied, due to the removal of non-crystalline material. The adsorption capacity the $N_2$ isotherms present with respect to those of $O_2$ are greater because the $N_2$ molecules present a greater quadrupole moment (0.31 $A^3$) than that of $O_2$ (0.10 $A^3$) (see Table 1). Therefore, they are more susceptible to the cationic content of adsorbents. In other words, the chemical treatments to which the zeolites are subjected cause an increasing accessibility towards the microporous network of crystalline aluminosilicates. The apparent discrepancies in the values reported in Table 2, obtained from the experimental adsorption-isotherms, can be attributed to the slow diffusion of the gases. This slow diffusion can be due to a blockage of the cations ($M^+$) by the residual material. These residues are being dislodged from the passages or entrance windows that communicate with channels A, B and C of these zeolites [5]. It can be expected that the adsorption

space is limited, and the difference in the size of the crystals plays a significant role in the estimation of the values obtained. In the equilibrium-pressure zone studied, there seems to be a good relationship between the experimental isotherms and the Langmuir model. The Henry constants ($K_H$) were obtained from the Langmuir equation, and in Table 3 the corresponding values are listed for the zeolites studied. In the eighth column, the values corresponding to the selectivity factor ($\alpha_0$) are listed and evaluated from the ratio of Henry's constants.

*4.6. Isosteric Enthalpy of Adsorption and Selectivity Coefficients ($\alpha_4$)*

Figure 7 shows the behavior of the isosteric enthalpies of adsorption for these gases as a function of the chemical treatment to which these zeolites were subjected.

From this figure, the behavior of the isosteric of adsorption for both gases follow a different behavior:

(1) the qst for $N_2$ decrease, depending on the chemical treatment;

(2) the qst for $O_2$ show a predominance of collateral interactions between the adsorbate molecules themselves over heterogeneous interactions. The first behavior is typical for materials that present heterogeneous interactions, energetically speaking. The values of qst for both gases are represented in Table 7. From this table you can see the following trend: qst $N_2$: H1 > H3 > H2 > H4 > N; while for qst $O_2$, H3 > H2 > H1 > H4 > N. For $\alpha 4$ the trend is H4 > H1 > H2 > H3 > N.

## 5. Conclusions

The conclusions of this investigation are the following:

1.  The results shown indicate that the acid-modification procedure of zeolitic samples yielded an increase in the intensity of the peak corresponding to quartz. This behavior can be related to the progressive dealumination that the crystalline structure suffered as the number of acid treatments increased, while quartz moieties remained mostly unaffected by the HCl leaching action.

2.  Acid treatment of these high-silica natural clinoptilolites has rendered improved adsorbents via the mechanism of decationation and dealumination, and also by dissolution of any amorphous silica blocking the channels A, B and C of the clinoptilolite structure. The dealumination process is favored to a greater extent in sample H2.

3.  The SEM of these zeolites shows us that the form of the obtained glasses of the CLINA zeolite presents a cubic symmetry with defined edges, and the crystals possess dimensions near to 2 μm.

4.  The N2 adsorption isotherms at 77 K on CLINA and H zeolites tells us the evolution of the shapes of the N2 isotherms with respect to the number of HCl washings. Porosity studies indicate the formation of adsorption isotherms of mixed type I-IV isotherms with hysteresis cycles of type H3, which are characteristic of lamellar pores for decationated clinoptilolite, while an isotherm type II-IV is characteristic of natural zeolites. The isotherms for the decationated zeolites show low-pressure hysteresis, which indicates an irreversible intercalation of molecules in pores of comparable size to the adsorbate molecules, or a pore-blocking effect.

5.  The separation of these gases (in pure form and in a mixture) was studied and confronted, based on different models, establishing that in some cases inconsistencies occur due to diffusion and the size of the crystallites. When evaluating the degree of interaction of N2 and O2 in pure form by means of the Clausius–Clapeyron relation, results which were like those previously reported in zeolites of these characteristics were found.

**Author Contributions:** All authors discussed and agreed upon the idea and made scientific contributions: writing—original draft preparation, M.Á.H.; experiment design, M.Á.H. and G.I.H., experiment performance, R.I.P. and J.D.S.-J.; data analysis, E.R.; writing—review and editing, M.d.l.Á.V. and V.P. All authors have read and agreed to the published version of the manuscript.

**Funding:** This research received no external funding.

**Acknowledgments:** This work was supported by the VIEP and the Academic Body "Investigación en zeolitas," CA-95 (PROMEP-SEP).

**Conflicts of Interest:** The authors declare no conflict of interest.

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
