# Peer review of "Influence of Chemical Pretreatment on the Adsorption of N2 and O2 in Ca-Clinoptilolite"

_separations, doi:10.3390/separations10020130_

Round 1

Reviewer 1 Report

1. The whole manuscript looks untidy, and not professionally written. The surname of one author is wrongly typed. Introduction must be rewritten to involve the readers into the main topic. There are too many unnecessary cited references (13 self-citations out of 33 references in total). The authors didn’t state the origin of the Figures 1 and 2, placed in the Introduction section; have they reused them or record during the reported experimental. Right sides of the Figures 1 and 2 are not clear in presentation, the yellow balls can’t be noticed. Actually, it looks like Fig. 1a is the same as Fig 4b.

2. “Earlier, we reported on the results of previous experimental studies for natural zeolites, such as XRD [22], SEM [23], EDS [24], HRADS [25];…”

It is not necessary to retell what the authors did before, in the experimental part of the work, if the results of the cited characterization do not refer to the materials used in the work.

3. Experimental section must be enriched with many details of the performed preparation procedure: “selected dealuminated clinoptilolite zeolites (CLIDA, H1, H2, H3, and H4).”

This is necessary because in the cited references detailed synthesis procedures are also missing. “preparation methodology was also published previously [26, 27].”

4. Experimental subsection 2.2. Methodology does not contain the methods discussed in the Results, such as XRD and SEM-EDS.

5. Results: In general, the materials are poorly characterized. SEM analysis of the modified samples is lacking, In the Table 2., there are some strange values, for example, in H1 the sum of all oxides is only 59.253 %, what are the remaining contents.

6. No loss of crystallinity was observed during dealumination process for CLINA zeolites, Figure 3. For which samples does the dealumination occur when according to the Table 2, the ratio of Si/Al is only higher for H H4 compared to the ratios of CLINA.

7. Authors cannot say in the Conclusion: “These zeolites were also characterized by XRD, EDS and HRADS using N2 at 77 K”, because in the manuscript the authors haven’t even defined what is HRADS (probably, high resolution adsorption).

Overall, I cannot recommend this manuscript to be accepted because it doesn’t meet the quality criteria of Separations. I hope that I have pointed out here some of the manuscripts faults to support the given conclusion.

Author Response

We would like to thank you for your attention to our article separations-2142399. Additionally, we want to express our gratitude to the reviewers for detailed reading, a friendly discussion, and suggestions that helped to improve the presentation of these results. We carefully revised the manuscript, therefore, corrections in the text are made in <  >.

Reviewer 2 Report

Dear authors,

You have presented an interesting research paper, but I think there is room for improvement in some details:

-You have not described how the HCl treatment is performed in the Methods section. It is difficult to understand the results without this information.

-Calculation Methods. Many of the symbols representing constants are not in the text or are incorrectly expressed (especially subscripts and superscripts). Please check this when creating the pdf file. This also happens in other parts of the text, e.g. the footnote of Table 4.

-The data in columns N2 and O2 of Table 5 are already in Table 4. It is not necessary to repeat them, and even less so with a change of name which can lead to confusion. Another option is to merge Tables 4 and 5 into one.

-The legend in Figure 7 is not fully visible.

-Line 304 "The H isotherm presents a very wide and open...". Do you mean one of the isotherms of the Hn samples or all of them in general? In either case, the sentence needs to be corrected.

Author Response

(The authors gave the same response as above.)

Reviewer 3 Report

The manuscript aims to estimate the degree of interaction of O2 nd N2 in natural clinoptilolite zeolite and the decationated with HCl. The manuscript needs a serious revision before it is considered to be published. A lack of intensive discussion is a weak point. The results are not interpreted appropriately. Introduction must be improved: relevant previous studies must be presented and there is no linkage between literature review and the objectives of the study. Conclusion looks liked a part of Discussion and it needs to be rewritten. The English language needs improvement. The specific comments are as following:

-       Identification of each peaks in the XRD result is required. Although the peak positions are similar, their intensities might be different. Authors might calculate the crystal size and discuss on this issue.

-       The changes in chemical compositions with HCl treatment should be discussed, rather than only reporting the order of each oxide (section 4.2). 

-       Micropore, mesopore, and total pore volumes should be reported in Table 3 and discussed along with the specific surface area and the results in Fig 6.

-       In Experimental section, the detail of the preparation of H1, H2, H3 and H4 is missing.

Author Response

(The authors gave the same response as above.)

Round 2

Reviewer 1 Report

The authors acted according to the remarks of the reviewers and thereby significantly improved the quality of the manuscript.

Reviewer 3 Report

The manuscript was corrected according to the comments